# Benchmarking General Purpose In-Context Learning

## Abstract

In-context learning (ICL) empowers generative models to address new tasks effectively and efficiently on the fly, without relying on any artificially crafted optimization techniques. In this paper, we study extending ICL to address a broader range of tasks with an extended learning horizon and higher improvement potential, namely General Purpose In-Context Learning (GPICL). To this end, we introduce two lightweight benchmarks specifically crafted to train and evaluate GPICL functionalities. Each benchmark encompasses a vast number of tasks characterized by significant task variance. These tasks are also crafted to promote long-horizon in-context learning through continuous generation and interaction, covering domains such as language modeling, decision-making, and world modeling. The benchmarks necessitate the models to leverage contexts and history interactions to enhance their capabilities, which we believe to be the key characteristics of GPICL. We present baseline solutions for the two benchmarks using transformer model and its variants, demonstrating that these benchmarks match the criteria for GPICL by highlighting the importance of long-term in-context dependencies and high potential for in-context improvement. Furthermore, our findings suggest that the scale of the parameter alone may not be the key factor for ICL or GPICL success; instead, greater importance should be given to higher task diversity and longer context lengths.

## 1 Introduction

The success of large-scale generative language models can be primarily attributed to two key factors: 1. *Zero-shot generalization* derived from the extensive accumulation and storage of knowledge within the model parameters during the pre-training and fine-tuning stages (Kojima et al., 2022); 2. *In-context learning*(ICL) to distill knowledge from the context during the inference stage (Brown et al., 2020). The ability of zero-shot generalization remains unchanged post-training, which parallel the concept of "innate abilities" (Koulakov et al., 2021) of biology. The ICL capability allows the agent to handle various unseen tasks without modifying the model parameters, only resulting in changes to the hidden states and memories. Currently, the ICL capabilities of large language models are confined to natural language-based tasks and relatively naive tasks such as following instructions and mimicking demonstrations. However, they generally lack the ability to continually interact with real-world environments and engage in reinforcement learning from contexts. Additionally, extremely complex tasks that require substantial amounts of context still pose significant challenges.

On the other hand, there are significant gap between artificial intelligence and biological intelligence, particularly regarding the functionality to continuously learn throughout their lifespans (Parisi et al., 2019). The enormous lifelong learning and adaptation potential of biological neural networks that forms abrupt contrast with their limited innate abilities at the initial stages (e.g., the stage of mammalian infancy) are also used to inspire the design of machine intelligence systems (Zador, 2019; Wang et al., 2022; Schmidgall et al., 2024). Among those a promising topic is general purpose in-context learning (GPICL) (Kirsch et al., 2022; 2023). GPICL involves meta-training across a wide range of task classes and learning to interpret in context to generalize to new tasks. However, the characteristic of diversity in this context can be challenging

---

source code will be available in the final version

| Methodology | Zero-shot Capability | ICL Potential | ICL Horizon |
|---|---|---|---|
| Multi-task Learning | Medium | Low | Short |
| Meta-Learning | Low | Medium | Short |
| ICL | High | Medium | Medium |
| GPICL | Low | High | Long |

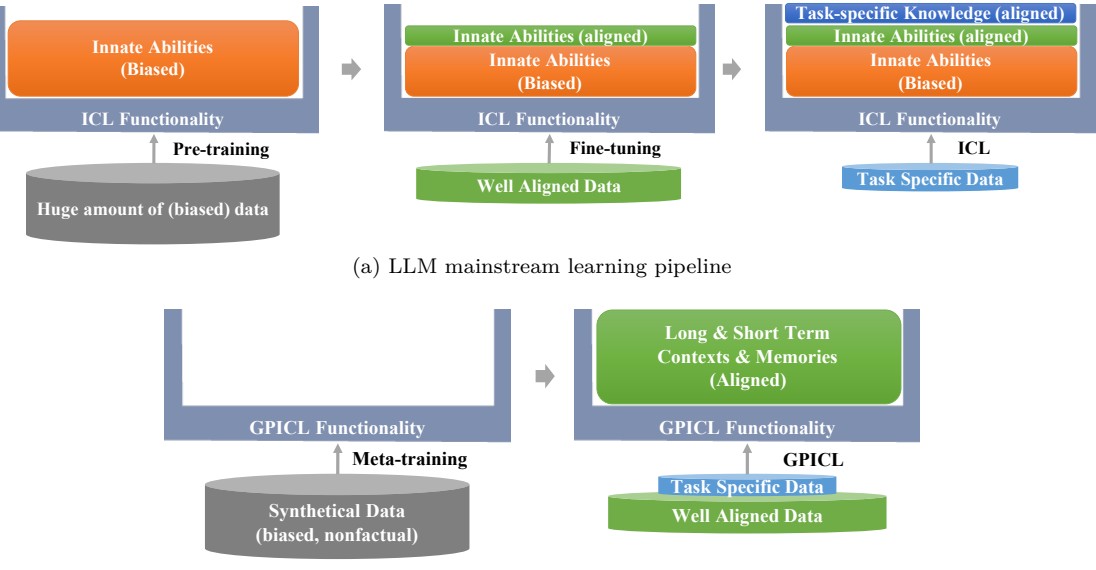

Figure 1: Conceptual diagram of in-context learning (ICL) and general purpose in-context learning (GPICL)

to quantify. We aim to enrich the concept of GPICL by incorporating additional features, as shown in Figure 1. We consider the meta-learning framework in which a model is meta-trained in the outer loop to utilize contextual information for task adaptation in the inner loop. We characterize a model with GPICL capability by three main features: 1. Low zero-shot performance, indicated by the model's performance when the context length is zero, i.e. no contextual information is available; 2. High in-context learning (ICL) potential, reflected by the gap between the asymptotic performance and the zero-shot performance; 3. A long ICL horizon, signified by the minimum context length necessary for the model to achieve convergence in performance improvement.

(a) LLM mainstream learning pipeline

(b) GPICL learning pipeline

Figure 2: Learning pipelines based on GPICL versus the current LLM learning pipeline (Ouyang et al., 2022).

**Importance of GPICL**. The objective of extending the in-context learning (ICL) horizon and potential is widely acknowledged. However, suppressing zero-shot performance seem to contradict the prevailing trend. To elucidate its importance further, we emphasize three key points:

First, theoretically, the lower zero-shot capability indicates a reduced reliance on inductive biases, thereby demonstrating sufficient diversity of meta-training tasks. As a result, the meta-training process focus primarily on the functionality of ICL instead of task-specific knowledge. Conversely, the attainment of zero-shot capability might suggest that the tasks possess a common knowledge base. While it could lead to faster adaption within the scope of these tasks, it might potentially restrict the model's applicability beyond the scope of these tasks.

Second, practically, meta-training the learning functionality requires an enormous number of tasks, which is challenging to collect in the real world. The use of abundant synthetic data in pre-training and meta-

training can be essential, but tend to be non-factual and biased, presenting challenges such as the sim-to-real gap. Ideally, the meta-training stage with the characteristics of GPICL results in the loss of most task-specific knowledge while preserving the ICL functionality, thus helps to prevent the model from the biases in the synthetic datasets. Building on this point, an alternative approach could be developed to replace the current mainstream learning pipeline of Pre-training & Fine-tuning & ICL with Meta-training & GPICL (Figure 2). In the new pipeline, initially, massive synthesized data, potentially containing a wealth of nonfactual and biased information, is used exclusively to cultivate the GPICL functionality. Subsequently, a considerable amount of high-quality, unbiased, well-aligned, real-world data is employed to enrich the contexts and memory, enabling the agent to achieve capabilities approaching those of humans.

Third, since a model's performance typically depends on both in-context learning (ICL) and zero-shot generalization, it is challenging to investigate and improve ICL independently. Therefore, we believe it is advantageous to construct benchmarks capable of eliminating the influence of zero-shot generalization and isolating the ICL capability, thereby facilitating the analysis of model architectures and learning algorithms.

**Guidelines for designing GPICL benchmarks**. We note that a major impediment to research in this area is the lack of benchmarks that adhere to the GPICL standard. We suggest that those benchmarks should satisfy the following standard:

- *Vast number of variant tasks with minimal inductive bias*: It is desirable to ensure that the meta-training and meta-testing tasks are sufficiently diverse, preventing the transfer of task-specific knowledge among different tasks. This guarantees that the model's performance is primarily driven by its in-context learning capabilities rather than inductive biases. It is also essential to require numerous tasks to prevent the model from hacking in-context learning (ICL) by merely memorizing all possible tasks and performing task identification, a limitation that many meta-learning benchmarks fall into when given limited tasks, as is discussed in Kirsch et al. (2022).

- *Interactive and generative tasks*: GPICL should include task exploration and completion processes that are both interactive and generative, to ensure obtaining complex functionalities such as in-context reinforcement learning. Consequently, benchmarks that solely focus on encoding a single sample characterized by extended contexts and features and conducting single-step predictions or classifications do not meet the criteria.

- *Lifelong in-context learning*: To guarantee the capability of adapting complex tasks, GPICL should handle very long-term dependencies that can scale up to millions or even billions of steps, effectively supporting the learning process over extensive periods (ICL horizon $T$ should be "from cradle to grave"). It is important to clarify that $T$ should denote the "minimum steps" possible given "ideal ICL functionality", excluding scenarios requiring extensive contexts that are primarily due to inefficient learning functionalities.

**Contribution of this paper**: We propose two critical benchmarks that adhere to the GPICL standard and are capable of generating unlimited synthetic data: 1. Meta-Language: We create a randomized yet consistent language pattern to meta-train models, demonstrating that a model, without prior exposure to natural language corpora, can acquire the ability to learn a new language from scratch through ICL; 2. Maze World: We generate an unlimited number of tasks within randomized mazes, where the model is meta-trained solely through context to learn exploration and navigation. For both benchmarks, we provide limited experiment results of baseline models based on Transformer (Vaswani et al., 2017) backbone and its variants. Moreover, we demonstrate how the number and diversity of tasks impact generalizability and contribute to GPICL. The primary goal of this paper is neither to introduce a new model or learning methodology, nor to provide an exhaustive survey of the existing ICL systems; rather, we aim to demonstrate through these baselines the potential for these benchmarks to benefit the research of ICL and GPICL. One crucial conclusion from these limited experiments may be that the scale of parameters alone is not crucial for ICL or GPICL. This finding aligns with previous research indicating that the size of contexts, and the scale of the memory states can play a more significant role (Kirsch et al., 2022; Wang et al., 2022) in learning from contexts.

## 2 Learning to learn an randomized language: Meta-Language

### 2.1 Problem Setting

Most large language models (LLMs) are trained on specific, existing languages. Rather than synthesizing existing languages, we primarily consider learning a new language as a task to focus more on the ICL functionality. Although there are hundreds of languages in the world, treating each language as a sample for GPICL provides only a few hundred samples, which is insufficient. We introduce the "Meta Language" benchmark. This benchmark is designed to generate a vast number of new "languages" and assess the models' ability to learn an arbitrary new "language." This pseudo-language is simulated using n-gram settings, generated by a randomized neural network $f$. The probability of the next token is determined by its preceding $n$ tokens and a set of random parameters $\theta$.

$$p(x_t|x_{t-1}, x_{t-2}, ...) = f(x_{t-1}, ..., x_{t-n}; \theta), \tag{1}$$

---

**Algorithm 1** Generating a length=T sequence of meta langauge

---

1: Randomly Sample $\theta \sim \mathcal{N}(0, \sigma^2)$, ($\theta$ is the parameter of the specified n-gram model $f$)
2: **for** t in [1, T] **do**
3:     $z_1, ..., z_N = f(x_{t-1}, ..., x_{t-n}; \theta)$
4:     For $i \in [1, N]$, $z_i = \lambda[z_i - \mathbb{E}(z)]/\sqrt{\mathbb{V}(z)}$
5:     $p(x_t = x_i) = exp(z_i)/\sum_j exp(z_j)$
6:     Randomly sample $x_t \sim p(x_t)$

---

The generation process is as simple as Algorithm 1. To maintain the diversity of the generated meta language, we also normalize the mean and variance of the logits input to Softmax to keep the distribution being neither too stiff nor too flat. We set the hyper-parameter $\lambda$ in Algorithm 1 such that the perplexity of the ground truth generator $\mathbb{E}[-logp(x_t))]$ is kept between 0.5 and 1.0. The sequences generated by the randomized generator are meaningless and chaotic. However, as long as the generator's parameters remain static, a model can progressively learn and capture the underlying rules of the randomized generator. The complexity of the generated sequences can be bounded by $N^n$, where $N$ is the size of the vocabulary. Therefore, with a sufficiently large $N$ and $n$—parameters that can be adjusted as needed—the complexity of the pseudo-language can be made arbitrarily high to test super-long-term dependencies.

### 2.2 Baseline Solution

As the training data, we generated 500K sequences with a pre-defined vocabulary size of $N = 32$, and the complexity of the pseudo-language was uniformly sampled across $n \in 3, 4, 5, 6$. Each sequence comprises $4,096$ tokens, totaling 2B tokens. We apply the auto-regressive transformer (Vaswani et al., 2017) with rotary embedding (Su et al., 2024). The hyper-parameter settings include: a tiny-sized transformer (303k parameters), a small-sized transformer (9.5M parameters), and a standard-sized(151m parameters) model. The evaluation set consists of independently sampled sequences, totaling 4K samples (16M tokens) for each $n \in \{2, 3, 4, 5, 6, 7, 8\}$, sampled uniformly. We assessed the model's performance by plotting the averaged perplexity $(-\log p(x_t))$ for each position across different $n$ values. To assess how the trained model generalizes, we also included the PG-19 (Rae et al., 2019) test set as a natural corpus. In this set, words are broken down into 26 alphabetic characters and randomly mapped onto 32 pre-defined tokens, the left slot are given to punctuation marks. The entire test texts are concatenated into a long sequence and broken down into sentences of 4,096 characters each. It is important to note that the model has never been trained on any natural language corpora. Details of the model description can be found in Appendix A.1.

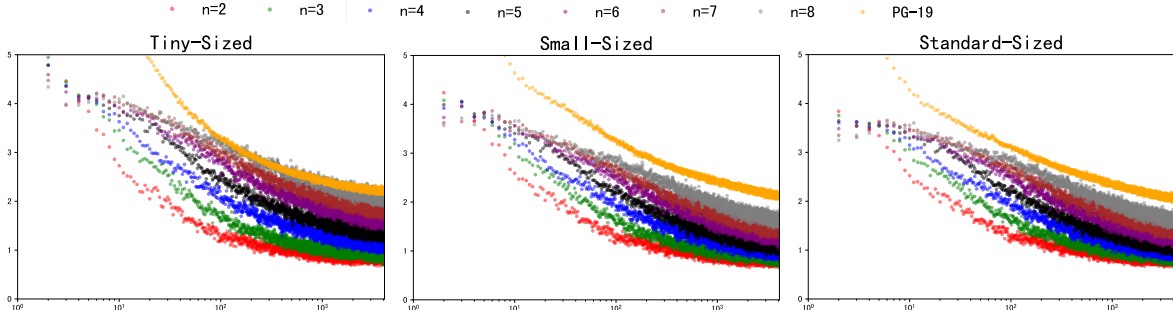

Figure 3: Comparing evaluations on meta-language sampled by different language complexity. Horizontal axis represent the position or context length, and vertical axis represent the perplexity (the lower the better).

## 2.3 Evaluation Results

### 2.3.1 Towards Meta-Language Model

We demonstrate that this pseudo-language trained model can be regarded as a **meta-language model** designed to adapt to any conceivable language through GPICL. See from basic evaluation results of 151M model on the test sets in Figure 3, several observations are noteworthy: First, perplexity improves with increasing context length across all values of $n$, indicating that the model is effectively leveraging in-context learning. Notably, for $n = 7$ and $n = 8$—complexities on which the model was not explicitly trained—we observed a consistent semi-logarithmic improvement in perplexity with increasing context length. Second, a higher language complexity, indicated by larger values of $n$, correlates with a more gradual improvement in perplexity relative to context length. This phenomenon confirms that increased language complexity introduces longer-term dependencies within contexts, establishing meta language as an ideal benchmark for evaluating long-dependency modeling capabilities. Most importantly, the model demonstrates a significant performance gain through in-context learning (ICL) in realistic natural language scenarios, despite never being exposed to a single real-world natural word, illustrating that the ICL capability acquired through meta-language is not restricted to a specific language type. We further test the ability of generation of the meta-language model on two rudimentary real-world language tasks, showing that in contrast to the absence of any natural language corpora, the model is able to learn to adapt to the real world language tasks through ICL only (Appendix A.4).

### 2.3.2 Scaling Laws for GPICL

We have conducted further research into how the scale of parameters affects GPICL capability. As illustrated in Fig. 4, our findings reveal notable trends: 1. Language complexity ($n$) influences the relative performance of different-sized models and affects the dependence on context length. In simpler tasks ($n < 4$), performance rapidly converges to lower bounds asymptotically ($t \sim T = 4k$. Here, with a slight abuse of notation, $T$ does not necessarily represent the ICL horizon, but rather denotes the maximum context length used in our experiments.), and the performances of various models become indistinguishable, despite starting differently. 2. It should be noted that for tasks as straightforward as $n = 2$, where a 300K model achieves comparable asymptotic performance to a 150M model, we observe that larger models cannot attain higher performance without sufficient context. It is reasonable to infer that increasing model complexity further will not be beneficial. This observation aligns with a key characteristic of GPICL, where learning from context takes precedence over zero-shot generalization. 3. For greater complexity ($n > 6$), the ICL horizon clearly extends far beyond $T = 4k$, and the improvement in perplexity exhibits polynomial scaling law (depicted as a linear relationship on a semi-logarithmic axis) with increasing context length. However, it remains difficult to assert from current experiments whether, given sufficiently long contexts, models of varying scales will converge to similar asymptotic performances. Nevertheless, we can confidently conclude that for $n < 8$, a model with 10 million parameters has been adequate. 4. For the PG-19 test set, we were surprised to find that the tiny-sized model achieves comparable asymptotic performance to the other two models. We must consider

that the complexity of natural language far exceeds that of any conceivable $n$-gram language. Furthermore, the improvement does not exhibit a linear trend on a semi-logarithmic scale. It is reasonable to hypothesize that the GPICL is only capturing certain short-term and high-frequency patterns in natural language (such as frequently appeared words).

Our experiments suggest a shift in the GPICL paradigm from increasing the parameters to increasing the context length, memory, and states provided the parameters are sufficient. In contrast, current large language models (LLMs) typically require at least billions of parameters, the majority of which we suspect are employed for zero-shot generalization, while the parameters dedicated to Incremental Context Learning (ICL) are relatively sparse. This perspective is consistent with the concept of the genomics bottleneck (Wang et al., 2022), which advocates for models that, while adequately parameterized, prioritize larger memory capacities to optimize GPICL. However, it is important to note that while the transformer architecture offers a "full memory" capability for sequence modeling, it also incurs a computational cost of $\Theta(T^2)$. This underscores the necessity for developing more efficient transformers, which would allow for the extension of context length to longer spans. We believe research in this area (Tay et al., 2022) could benefit from this benchmark.

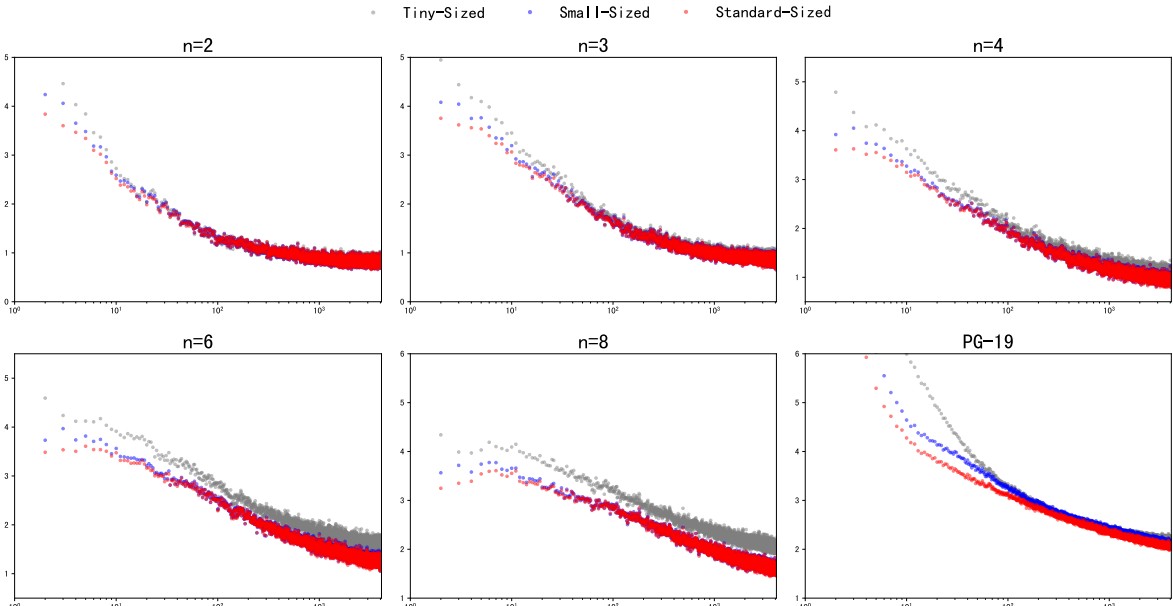

Figure 4: Comparing performances of language models of different sizes on the the same test set. Horizontal axis represent the position or context length, and vertical axis represent the perplexity (the lower the better).

## 3 Learning to navigate an randomized world: Maze World

### 3.1 Problem Setting

Mazes are typical environments that can scale up to an unlimited number of layouts, features, targets, and tasks. Many previous works benchmark meta-learning with mazes, but these are limited to 2-D observation space (Wang et al., 2022; Morad et al., 2023; Nikulin et al., 2023), text-only (Ding et al., 2024) observation and action space, and limited number of layouts and tasks (Duan et al., 2016; Morad et al., 2023; Ding et al., 2024). To enhance the task diversity, we propose the Maze World benchmark. It tasks an agent with navigating a completely unknown environment, requiring it to explore, memorize its surroundings, perform simultaneous localization and mapping (SLAM), identify navigation targets, and plan its route. This setting could potentially reflect a wide range of real-world tasks, such as household robots adapting to unfamiliar indoor environments. As the agent gains more experience within a specific environment, it should progressively gather more information and navigate to specified targets more efficiently. The proposed benchmarks has the following features:

- Each task corresponds to a randomly generated $n \times n$ maze, with layouts and textures assigned randomly.

- A number of Potential Navigation Targets (PNTs) are placed at random locations and remain fixed throughout the task. Each PNT is marked by a semi-transparent beam of light, as shown in Figure 5(a).

- For each task, a randomized sequence of commands is generated, each directing the agent to walk to a specified target that is identifiable by color.

The agent receives a positive reward for reaching the specified target, while a minor step-cost is incurred for each step taken. Available actions include moving forward, moving backward, stopping, turning left, and turning right. Note that unlike previous works (Chen et al., 2021; Kirsch et al., 2023; Lin et al., 2023; Lee et al., 2024), we did not use rewards as inputs, nor did we explicitly predict rewards. In the "NAVIGATION" tasks of Maze World, instant rewards are easily inferred from observations. However, in other tasks within Maze World, such as "SURVIVAL" tasks (see details in the Appendix A.6), modeling rewards may be necessary. This should not impede the analysis of the benchmark and GPICL process.

To provide a cost-effective and scalable reference dataset for meta-training GPICL, we have implemented rule-based agents with a robust simulated policy. These agents are allowed to access the global maps, but with limitations: They are allowed to record map areas within their line of sight into memory, while other areas remain hidden. These agents are referred to as *privileged agents* because they have access to additional information. The privileged agents efficiently navigates to their goals by utilizing an exploration-then-exploitation strategy. Although these agents may not consistently reach the global optimum due to shortcomings in its exploration strategy, it provides a relatively high standard reference policy for imitation learning.

To simulate variant levels of intelligent agents, we equip the privileged agents with both Long-Term Memory (LTM) and Short-Term Memory (STM). The STM retains only the most recent three pieces of observation, while the LTM retains information until the episode concludes. Only part of the STM is allowed to be recorded to LTM. We introduce a hyper-parameter, $p(\text{STM} \rightarrow \text{LTM})$, to govern the probability of transferring information from STM to LTM. At $p(\text{STM} \rightarrow \text{LTM}) = 0\%$, the agent operates without LTM, whereas $p(\text{STM} \rightarrow \text{LTM}) = 100\%$ allows for a fully functional LTM. For simplicity, a privileged agent with $p(\text{STM} \rightarrow \text{LTM}) = p$ is written as preiviledged agent (p) for short.

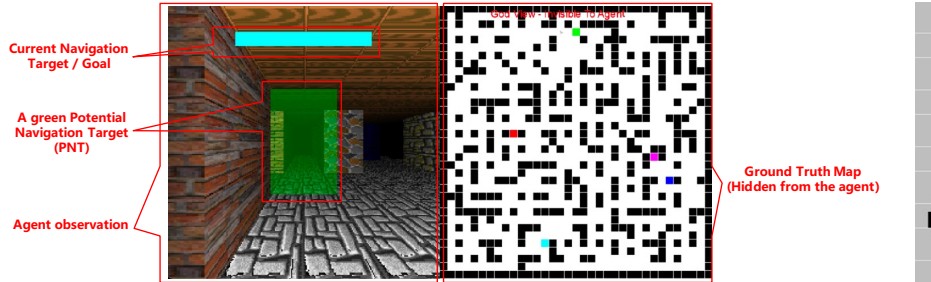
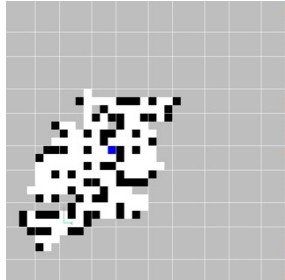

(a) A demonstration of observations and a global map in MazeWorld

(b) The LTM of a privileged agent

Figure 5: A demonstration of the maze world tasks alongside privileged agents with smart navigation policy. The privileged agent is provided with limited access to the environment's ground truth (b) under specific conditions. Only the areas within the agent's line of sight are revealed, while the unsighted regions remain obscured. The agent retains the visual information from the nearest three frames in its short-term memory, which is then transferred to long-term memory based on a specified probability. The privileged agent employs an exploration-then-exploitation strategy.

### 3.2 Baseline Solution

The baseline solution for the Maze World tasks is inspired by model-based imitation learning (Hu et al., 2022) and causal decision models (Chen et al., 2021). Although running meta-reinforcement learning (MetaRL) could be beneficial, we did not directly apply RL due to the significant challenges in scaling RL to meet the required needs. Following previous work that suggests imitating an expert as adequate to generate reinforcement learning capabilities (Lin et al., 2023; Zisman et al., 2023; Lee et al., 2024), we mainly focus on imitation learning augmented with world models. We use Variational Auto Encoder (VAE) to process each image to latent representation, and use a causal transformer (Chen et al., 2021) to encode the historical observation and action. The model ouputs the distribution of actions, and also the expected next frame. Specifically, given the reference trajectory $(i_1, a_1, i_2, a_2, ..., i_T, a_T, i_{T+1})$, where $i_t$ and $a_t$ represent the observation and action respectively, our model is represented by:

$$z_t = \mathbf{ENC}_{\text{vae}}(i_t), \tag{2}$$

$$\hat{i}_t = \mathbf{DEC}_{\text{vae}}(z_t), \tag{3}$$

$$\pi(a_0), \hat{z}_1, \pi(a_1), \hat{z}_2, ... = \mathbf{CAUSAL}(z_0, a_0, z_1, a_1, ...), \tag{4}$$

where $z_t$ is the latent embedding of the observed image $i_t$. Predicting the distribution of actions ($\pi$) and the next frame ($\hat{z}_t$) are typically referred as *Policy Model* and *World Model* respectively.

The learning losses are composed of three parts: 1. the reconstruction error ($\mathcal{L}_{vae}$); 2. the cross entropy between the reference action and the predicted distribution ($\mathcal{L}_{pm}$); 3. the next-frame prediction error ($\mathcal{L}_{wm}$). The final loss would be:

$$\mathcal{L}_{vae} = \sum_t ||\hat{i}_t - i_t||^2 + \lambda \text{KL}(p(z_t)||\mathcal{N}(0, 1)) \tag{5}$$

$$\mathcal{L}_{pm} = \sum_t -log\pi(a_t) \tag{6}$$

$$\mathcal{L}_{wm} = \sum_t ||\mathbf{DEC}_{\text{vae}}(\hat{z}_t) - i_t||^2 \tag{7}$$

$$\mathcal{L} = \alpha_{vae}\mathcal{L}_{vae} + \alpha_{pm}\mathcal{L}_{pm} + \alpha_{wm}\mathcal{L}_{wm} \tag{8}$$

The choice of causal transformer includes a small-sized transformer (26M parameters) and a standard-sized transformer (237M parameters). A overview of the model architecture is in Fig. 6. To train our baseline models, we collect training data include 120K episodes of demonstration from demonstration of privileged agents for imitation learning. Each episode has $2,048$ steps, which gives 240M frames in total. To avoid the growing "compounding errors" (Ross and Bagnell, 2010) associated with direct behavior cloning, we drew inspiration from dataset aggregation (DAgger) methods (Ross et al., 2011). We collected the trajectory of states by running a very noisy behavior policy, but used the privileged agent (100%) to label its action at each step (which is not actually executed). Additionally, to enhance the stability of auto-regression, we dropout and randomize 15% of the input observations to the causal transformer during training (Details in Appendix A.3).

To investigate the impact of context length on performance, we introduce additional two variants that share the same structure as the small-sized causal transformer (with 26M parameters), but have limited access to the contexts. The *partial-context* causal transformer has an attention window of 2, providing effective context length up to 9. The *context-free* transformer has an attention window of 1, providing effective context length of 1. The context-free and partial-context transformers are trained by further fine-tuning on the foundation of the full-context version (Appendix A.2).

### 3.3 Baseline Performances

**Static Evaluation**. We first conduct static evaluation using off-policy data from an independently sampled validation set comprising 1275 sequences (totaling 2.6M frames). To analyze the impact of context length on performance, we assessed the losses of the world model and policy model for each individual time step

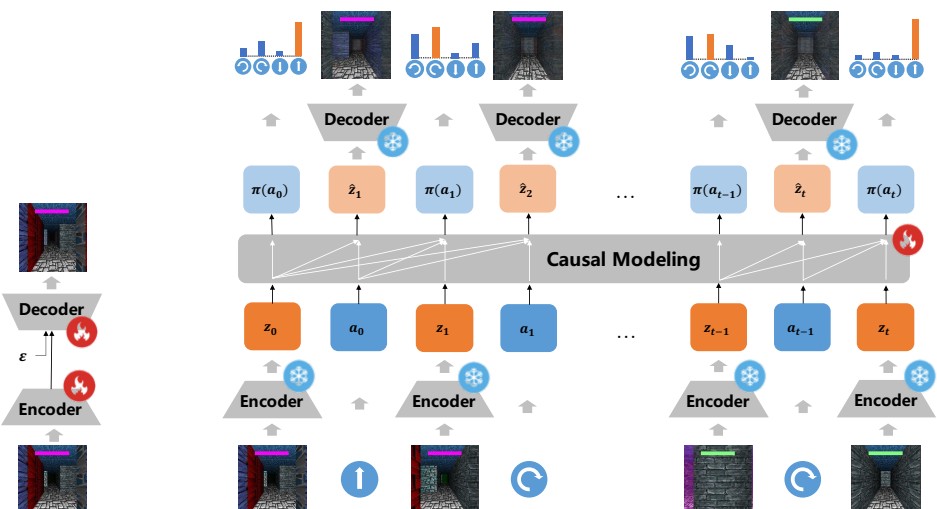

Figure 6: An overview of the model evaluated on the maze world tasks. Initially, a Variational Auto-Encoder (VAE) is trained to encode observed images into hidden vectors and vice versa, with its parameters subsequently frozen. The sequence of observations and actions is then utilized to train a causal model, specifically a transformer that employs only backward attention, using imitation learning and self-supervised learning techniques. For context-free and partial-context solutions, we intentionally mask some of the backward connections.

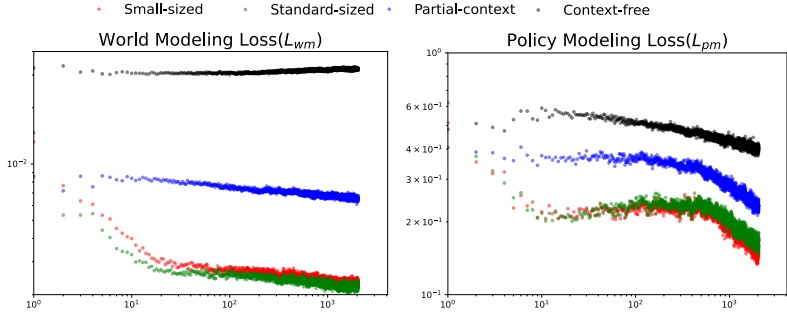

Figure 7: Static assessment of the ability to imitate previleged agents ($\mathcal{L}_{pm}$) and predict the next frame ($\mathcal{L}_{wm}$) across each time step ($t$).

($\mathcal{L}_{wm}(t)$ and $\mathcal{L}_{pm}(t)$). In Figure 7, We compare the performance of the four variants, with several noteworthy observations: First, the ability to encode long contexts significantly influences performance, while the scale of the parameters shows no observable effect, which is also consistent with the conclusion of Meta-Language. Second, both world-modeling and policy-modeling improve with increased context length, though they exhibit some subtle differences. The capability of world-modeling improves rapidly within the first 20 steps, after which the rate of improvement slows. This could be because single-step world modeling is primarily impacted by the localized environment, explaining the rapid initial improvement with short contexts, but is less influenced by distant layouts, leading to slower improvements over longer contexts. In contrast, the improvement in policy modeling with increased context length is not monotonic, with the loss remaining relatively unchanged between 10 and 500 steps. The decision modeling loss is closely related to the reference policy, which follows an exploration-then-exploitation strategy. These results may indicate that imitating the exploration strategy is more challenging.

**Interactive Evaluation**. It is more beneficial to investigate the online performance of the policy model and world model by directly interacting with the environment. We first directly evaluate the performance of the policy model by interacting with a set of evaluation tasks. The evaluation set includes 64 tasks across

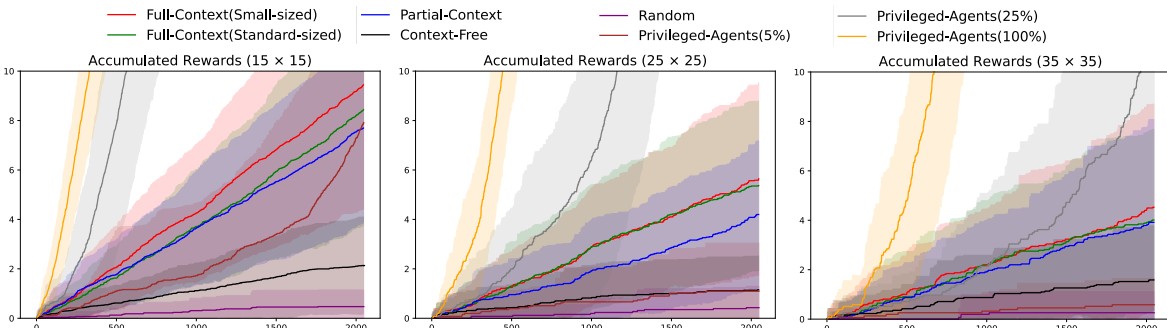

Figure 8: Interactive evaluation results of the randomized, previleged and casual-modeling based agents, depicted by plotting accumulated rewards against simulated time steps. The step minor cost is set to 0. Each plot summarizes the results from static 64 tasks. The shaded area represents the 95% confidence interval of the expectation.

mazes of sizes $15 \times 15$, $25 \times 25$, $35 \times 35$. We plot the average accumulated rewards and their confidence over time (t). For comparative analysis, we also include the performance of a random policy, where actions are randomly sampled regardless of the observations, as well as privileged agents with $p(\text{STM} \to \text{LTM})$ of 5%, 25%, 100%. The results (Figure 8) indicate that the model with full contexts outperforms the random policy, context-free agents, and privileged agents (5%), but there is a significant gap compared to privileged agents with 25% and 100% LTM. However, when compared with partial-context causal transformers, the superiority of the full-context model is not obvious. Considering the conclusion in static evaluation, it is reasonable to consider that the performance of interactive evaluation is constrained by imitation learning itself. We believe it is beneficial to incorporate on-policy trajectories in the dataset in the future, either in reinforcement learning or DAgger (Ross et al., 2011).

**In-Context Improvement of World Modeling**. In Figure 9, the world model is evaluated by comparing the ground truth future frames and predicted future frames at steps 1, 100, 1000, and 2000 of each interactive run. We demonstrate the performance of world models in predicting both the immediate future ($k = 1$) and a slightly more distant future ($k = 4$). The latter is more reliant on long-term memory, making it inherently more difficult. In the case of smaller $15 \times 15$ mazes, we notice a substantial enhancement in world modeling as the context length expands from 1 to 1000, which is validated by a consistent reduction of prediction errors for both $k = 1$ and $k = 4$. For larger-scale mazes, the challenge of predicting $k = 4$ is amplified, as retaining the entire map in memory becomes increasingly challenging. On the other hand, the partial-context model does not enhance its predictive capabilities as context length increases over 100, which aligns with expectations that it can not keep a long-term memory. The results indicate that the full-context causal transformer demonstrates a reasonable capacity to utilize contextual information for world modeling, yet there is considerable room for improvement, especially in solving larger-scale mazes and the efficient utilization of long-term dependencies. For further insights and cases, refer to Appendix A.5.

## 4 Emergence of Generalizability

To illustrate the relationship between the two benchmarks and GPICL, we delve into how the ICL capability is linked to the diversity of meta-training data in these benchmarks. In this section, we pre-select a set of tasks to generate the sequences (dataset) for meta-training. Unlike the procedural generation of tasks for each sequence, this approach reduces the diversity of both the task-set and the dataset, by allowing multiple sequences to be derived from a single task. For example, in Meta-Language, each task is defined by the parameters of the n-gram generator ($\theta$), allowing for the generation of different sequences from a single task through softmax sampling. Similarly, in Maze World, each task is characterized by the layout, textures, and the position of the PNTs (potential navigation targets), with different trajectories being generated by sampling behavior policies. To examine the effect of task diversity on ICL capability, we create datasets of the same size but with varying numbers of pre-defined tasks. In Meta-Language, we generate 500,000

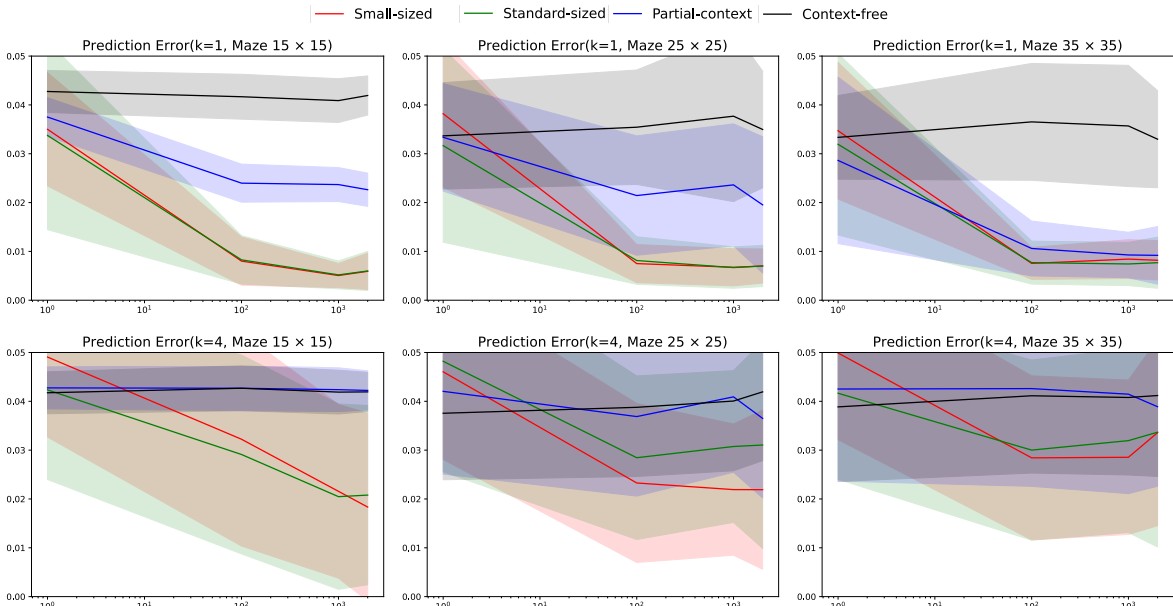

Figure 9: Results regarding the in-context improvement of world modeling in Maze World. We assess the mean square error (MSE) for forecasting subsequent frames. When $k = 1$, it indicates the error associated with predicting the immediately following frame. When $k = 4$, it denotes the cumulative error for predicting 4 frames ahead in an autoregressive manner. The predictions are initiated from various context lengths (or time step), with $t$ taking values from the set $\{1, 100, 1000, 2000\}$. The mean square error is averaged across 64 pre-selected tasks, and the shaded area provides a 95% confidence interval for these measurements.

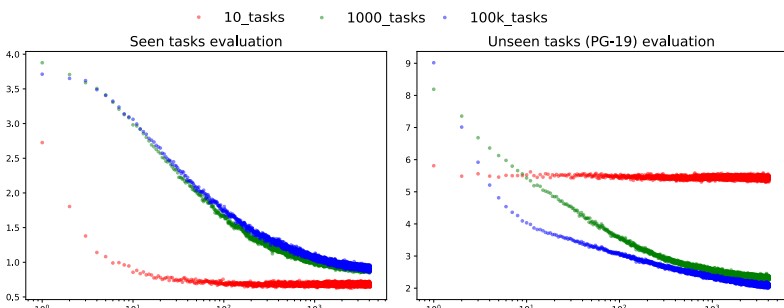

Figure 10: Performance of the meta-language model trained with 500,000 sequences generated from varying numbers of pre-selected tasks. The vertical axis represents the average perplexity in evaluation (lower values indicate better performance), and the horizontal axis represents the context length. Each point in the seen tasks evaluation is the average of 4,000 sequences generated from the pre-selected tasks. Each point in the unseen tasks evaluation is based on the PG-19 test set.

sequences (2 billion tokens) using 10, 1,000, and 100,000 pre-sampled tasks, respectively. In Maze-World, we synthesize 64,000 sequences (126 million frames) with 1, 10, 100, and 10,000 pre-sampled tasks for the $15 \times 15$ mazes. Given that the performance of standard-sized models and small-sized models was nearly indistinguishable in our previous experiments, we use only small-sized models for both benchmarks. Each meta-trained model is assessed on both 'seen' tasks (those used to generate the meta-training data) and 'unseen' tasks. The outcomes are displayed in Figures 10 and 11.

Several conclusions have been validated across both benchmarks: First, we observe a continuously increasing asymptotic performance (the performance with sufficiently long context) in unseen tasks as the number of meta-training tasks increases, a phenomenon also noted by Kirsch and Schmidhuber (2021). This finding

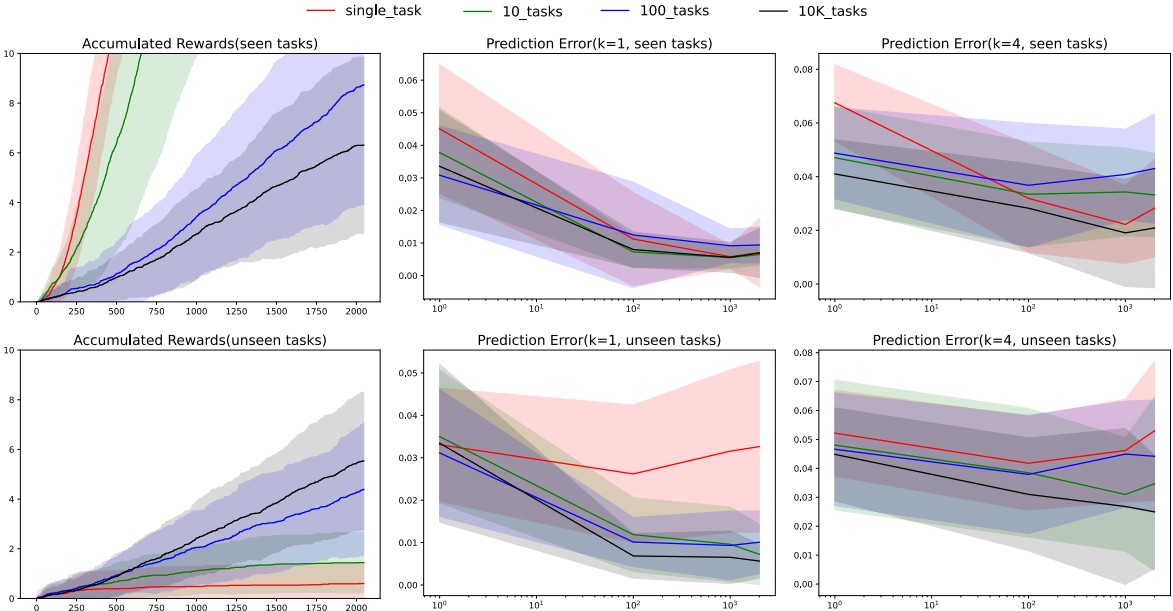

Figure 11: Performance of the causal transformer model trained with 64,000 Maze World sequences generated from varying numbers of pre-defined tasks. The left vertical axis represents the accumulated reward (higher values indicate better performance), while the right vertical axis represents the prediction error of world modeling in the interactive evaluation (lower values indicate better performance). The horizontal axis represents the context length or time steps. For the prediction error of world modeling, we evaluated the mean squared error (MSE) for predicting 1 and 4 steps into the future at context lengths of 1, 100, 1000, and 2000. Each evaluation point is the average of 64 runs, with the shaded area representing the 95% confidence interval.

confirms that the capability for task generalization is enhanced. Second, which is less discussed in the previous work, for seen tasks, performance does not consistently improve with the increase in the number of tasks; In contrast, it declines in most of the groups. Additionally, we notice an expansion of the ICL horizon, identified by the length of context required in the performance improvement phase, and a reduction in zero-shot performance, identified by the level of performance at the first time step (where context length is 0), in both seen and unseen tasks. We hypothesize that this may be a consequence of the model transitioning from *task-identification* to *learning-to-learn*. In this process, the model retains less specific information about each task but gains improved generalizability and the ability to learn through ICL. The decline in performance for seen tasks could also be attributed to the insufficient context length we employ (up to 2,048 and 4,096) and the size of the dataset. We deduce that, given sufficiently long context and large meta-training datasets: 1. In the evaluation of seen tasks, the asymptotic performance of groups with high task diversity can be comparable to that of groups with low task diversity. 2. In the evaluation of unseen tasks, the performance of groups with high task diversity should approach the performance levels observed in seen tasks.

In summary, it is important to note that the observed increase in generalizability, the expansion of the ICL horizon, and the reduction in zero-shot capability are all in line with the definition of GPICL that we previously mentioned. It is also worth to mention that we observe a slower convergence in meta-training as the number of tasks increases, despite the dataset size remaining constant (See Figure 15 in Appendix A.2). This suggests that the process of learning to learn is more challenging than mastering specific tasks.

# 5 Related Work

## 5.1 Meta-learning and GPICL

Meta-learning encompasses a wide range of methodologies, including gradient-based adaptation (Finn et al., 2017), attention-based adaptation (Mishra et al., 2018), and memory-based adaptation (Santoro et al., 2016; Duan et al., 2016; Wang et al., 2022; Lu et al., 2024). Both attention-based and memory-based adaptations are closely related to in-context learning, especially general-purpose ICL. Since the discovery that large-scale language models (LLMs) can perform few-shot learning (Ouyang et al., 2022), Transformer architectures and LLMs are increasingly becoming the state-of-the-art for meta-learning and in-context learning (ICL). This includes applications ranging from few-shot supervised learning (Min et al., 2021; Garg et al., 2022; Dai et al., 2023; Li et al., 2023), in-context reinforcement learning (Chen et al., 2021; Laskin et al., 2022; Lin et al., 2023), to multi-modal reasoning and embodied control (Reed et al., 2022; Szot et al., 2023). However, typical pre-training models only address relatively simple few-shot learning tasks, while more complex tasks still require task-specific or class-specific training. It is desirable to introduce general-purpose in-context learning (Kirsch et al., 2022; 2023) for generalizing across a wider variety of tasks, which necessitates benchmarks that can scale up in both quantity and diversity.

## 5.2 Relative Benchmarks

**Meta-Learning Benchmarks**. A variety of benchmarks have been utilized to evaluate meta-learning algorithms, encompassing domains such as image classification (LeCun et al., 1998), video games, and robot manipulation (Brockman et al., 2016). Many of these benchmarks were not initially designed for meta-learning but have been adapted to accommodate varying tasks through parametric adjustments in hyperparameters and by artificially hiding certain parameters from observation. This can be further advanced to include precedurely generated random targets of control (Finn et al., 2017; Mishra et al., 2018; Yu et al., 2020; Morad et al., 2023), video games of different difficulty level (Cobbe et al., 2019), geometry of locomotion (Najarro and Risi, 2020), rules of transition (Nikulin et al., 2023), and layouts of mazes (Mishra et al., 2018; Morad et al., 2023). However, some of these hidden configurations are relatively simple and fail to generate sufficient diversity, causing models to lean towards acquiring zero-shot and task identification abilities rather than in-context learning from scratch. For example, while many previous benchmarks include hiding the target of locomotion, an ideal learning algorithm should be able to locate the hidden target in at most three steps by comparing the rewards of walking in different directions, given that the location of the target is only in two dimensions. Another promising approach is to randomize labels (Kirsch et al., 2022; Wei et al., 2023), observations (Morad et al., 2023), and actions (Sinii et al., 2023), which tend to have higher dimensions. Based on the prior studies, it is plausible to infer that the diversity of tasks and the resulting generalization capabilities are related to the number of hyperparameters that can be varied. In contrast to the artificial randomization of rewards, observations, and targets, we believe that Meta-Language and MazeWorld offer environments that are inherently more complex and significant for benchmarking GPICL.

**Language Modeling**. Recently, emphasis has been placed on long-term dependency modeling in large language models (LLMs). Most of the newly proposed benchmarks support context lengths over 8K tokens (Rae et al., 2019; Tay et al., 2020; Bai et al., 2023; Shen et al., 2023; Song et al., 2024). However, there are several limitations when using current language-based benchmarks for in-context learning (ICL) investigation. First, some benchmarks focus on one-step classification with very long feature descriptions (Tay et al., 2020; Chalkidis et al., 2019; Bai et al., 2023), which is not suitable for examining the interactive performance of the model. Second, it is unclear how much long-term dependency actually exists in these long contexts (Chen et al., 2024). Even if some long-term dependencies are present, they may be weak and relatively simplistic (such as retrieval and copy tasks), and could be overshadowed by the powerful zero-shot generalization capabilities of the models. This overshadowing can obscure crucial information important for GPICL. In practice, it is often found that scaling up model parameters to enhance zero-shot generalization typically yields better results than extending the ICL horizon.

**Visual Navigation**. There are a few datasets featuring indoor and outdoor robot-environment interaction scenes, reconstructed from images taken in real-world settings (Song et al., 2017; Chang et al., 2017; Xia

et al., 2018), or simulated environments (Szot et al., 2021). Currently, most of these environments primarily facilitate the learning of zero-shot generalization. A significant limitation of these datasets is the high data collection cost and the challenge of scaling up. Compared to these benchmarks, the proposed Maze World has a much larger sim-to-real gap, but offers greater diversity and complexity, and is much easier to scale up. Additionally, we suggest that reality-generated data could better serve as contexts in the GPICL stage, as illustrated in Figure 2(b), rather than during meta-training.

## 6 Conclusions

In this paper, we introduce two benchmarks specifically designed for GPICL. Our preliminary investigations of these benchmarks have yielded inspiring results, demonstrating that scaling laws might exist between context length and performances, and that the performance is not entirely dependent on the scaling of parameters.

It is important to note that although we are seeing the potential of application of those datasets in areas including language modeling and indoor navigation, the synthetic data generated by those benchmarks are not yet validated to be ready for a direct help to application. This work can be expanded along two directions: 1. The development of more realistic GPICL benchmarks, for instance, by adding greater complexity to the meta-language; 2. Utilizing the benchmarks to uncover more sophisticated models (including memory-augmented models) and optimization techniques (such as reinforcement learning) to further extend the GPICL horizon and enhance performance gains.

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

# A  Appendix

## A.1  Model Description

Table 1: Comprehensive configurations of the baseline models

| Model | parameters | layers | $d_{model}$ | $n_{heads}$ | batch size (tokens) |
|---|---|---|---|---|---|
| Meta-Language Model (tiny) | 303K | 6 | 64 | 4 | 32K |
| Meta-Language Model (small) | 9.5M | 12 | 256 | 8 | 32K |
| Meta-Langauge Model (standard) | 151M | 12 | 1,024 | 16 | 64K |
| Causal Transformer (small) | 26M | 8 | 512 | 8 | 32K |
| Causal Transformer (standard) | 237M | 12 | 1,280 | 16 | 32K |

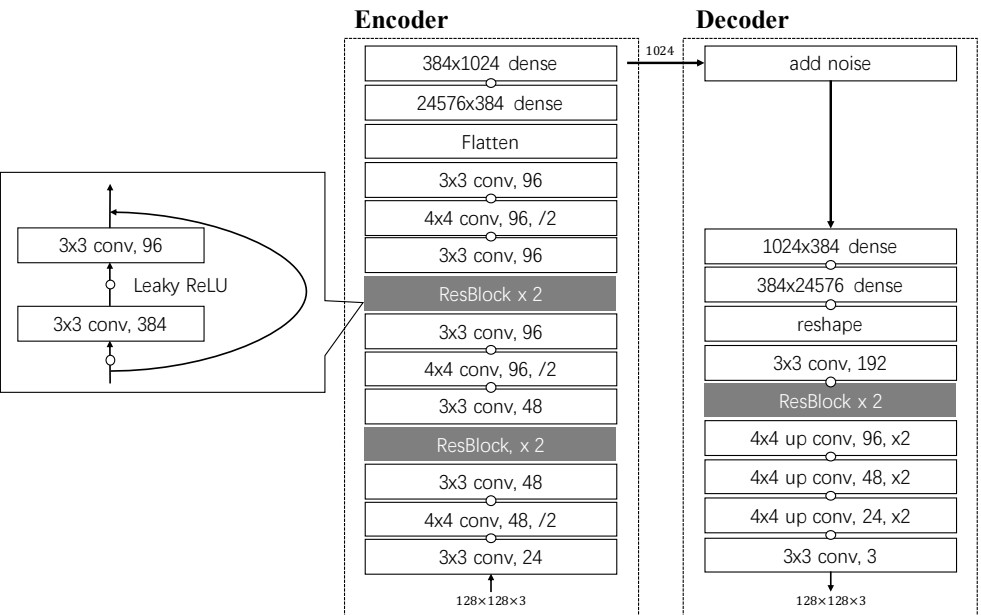

Figure 12: Structures of the encoder and decoder in the Variational Autoencoder

The comprehensive configurations of the baseline models are outlined in Table 1. Notably, within the Maze World framework, each temporal increment is encoded by a pair of tokens: an observation and its corresponding action. For the optimization of the Meta-Language and Maze World baseline models, we employed the Noam decay scheduler (Vaswani et al., 2017). This scheduler peaks at a learning rate of $10^{-3}$ following an initial warm-up phase spanning $1,000$ steps.

For the image encoder (11M parameters) and image decoder (14M parameters), we employ convolutional and deconvolutional layers augmented with ResNet blocks (He et al., 2016). The detailed structure of the VAE is depicted in Figure 12. Initially, we train the VAE in isolation by setting $\alpha_{wm}$ and $\alpha_{pm}$ in Equation 8 to 0. Subsequently, we freeze the parameters of the encoder and decoder and train the causal transformer only.

For partial-context and context-free causal transformers, we modify the attention window to 2 and 1, respectively, as shown in Figure 13. Since transformers can propagate information between layers, an attention window of 2 results in an effective context size of the number of layers plus 1.

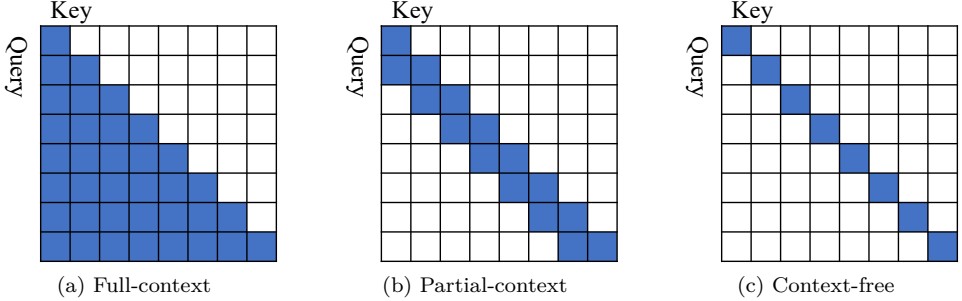

(a) Full-context      (b) Partial-context      (c) Context-free

Figure 13: Variant attention masks of causal modeling

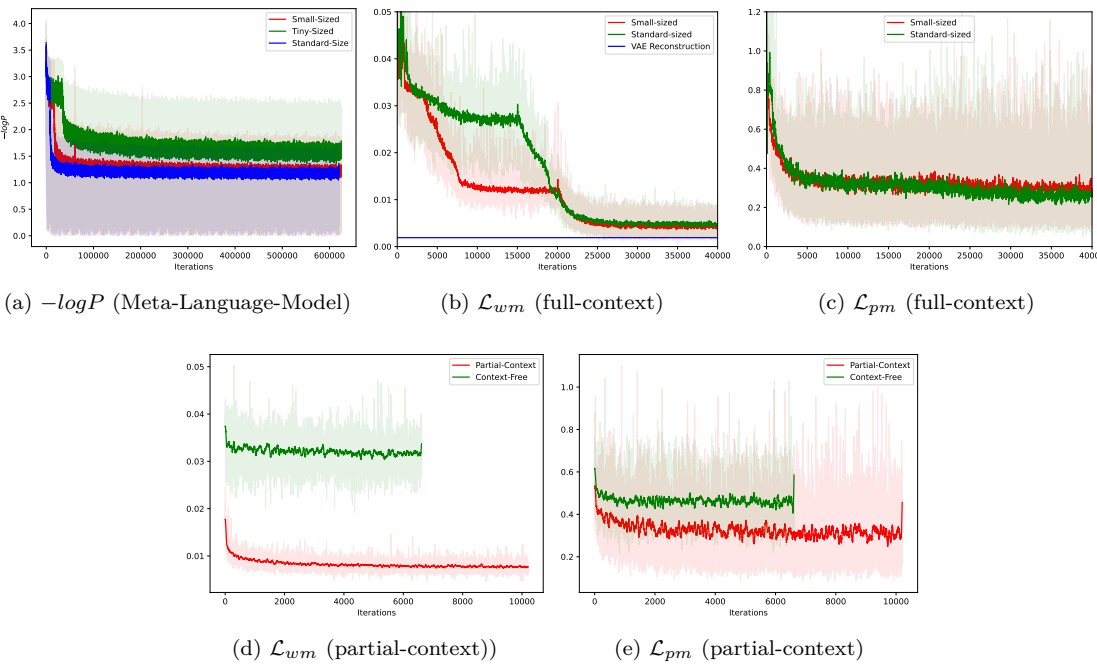

(a) $-logP$ (Meta-Language-Model)  (b) $\mathcal{L}_{wm}$ (full-context)  (c) $\mathcal{L}_{pm}$ (full-context)

(d) $\mathcal{L}_{wm}$ (partial-context))  (e) $\mathcal{L}_{pm}$ (partial-context)

Figure 14: The training loss with respect to the steps of optimization in training with procedural generated tasks and sequences. Notice that the partial-context and context-free causal transformers are initialized from the trained full-context transformers.

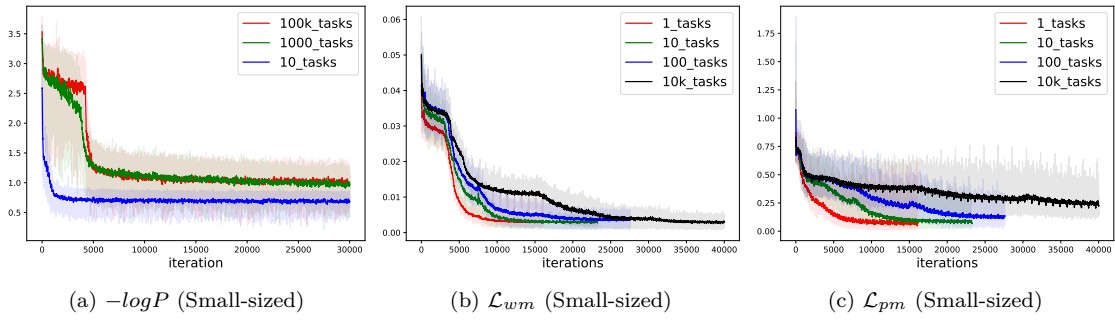

(a) $-logP$ (Small-sized)  (b) $\mathcal{L}_{wm}$ (Small-sized)  (c) $\mathcal{L}_{pm}$ (Small-sized)

Figure 15: The training loss with respect to the steps of optimization in training with sequences generated from pre-selected tasks.

## A.2 Convergence of Meta-Training

Figure 14 presents the training loss versus the number of iterations for training the baseline in section 2 and A.6. During training baselines for Meta-Language tasks, we observed that standard-sized models converge relatively slowly on the training set. To address this, we conducted additional warm-up epochs on a simpler pre-training dataset with $n = 2, 3$ before transitioning to the main pre-training dataset. During training the baselines for Maze World tasks, we found there was mutual interference between the world modeling loss ($\mathcal{L}_{wm}$) and policy modeling loss ($\mathcal{L}_{pm}$), especially in the initial stages. Our meta-training underwent several phases where we continuously adjusted the loss weights $\alpha_{wm}$ and $\alpha_{pm}$ in Equation 8. To further accelerate and stabilize the convergence of the world model, in addition to these losses, we introduced an auxiliary loss $\mathcal{L}_z = \sum_t ||\hat{z}_t - z_t||^2$ with a weight of $\alpha_z = 0.05$. During the final stage, the hyperparameters are adjusted to $\alpha_{pm} = 0.25$ and $\alpha_{wm} = 0.70$. However, we recommend starting with $\alpha_{wm} > 0.90$ and $\alpha_{pm} < 0.10$ as the

policy model training loss is more volatile initially. The context-free and partial-context causal transformers are initialized from the full-context causal transformer, and the hyperparameters remain unchanged.

Figure 15 presents the training loss versus the number of iterations for the experiments in section 4. Both group are experimented on small-sized models.

### A.3 Data Collection Strategy for Maze World

For dataset collection and static evaluation of Maze World, inspired by dataset aggregation (Ross et al., 2011) and noise distillation (Zisman et al., 2023), we utilize distinct behavior and reference policies. The behavior policy is determined by previleged agent where $p(\text{STM} \to \text{LTM})$ randomly sampled between $[0, 50\%]$. Additionally, with the probability of $\epsilon \in [0, 80\%]$ the behavior action is determined by random decision. With the states generated by behavior policy, we use the reference policy with $p(\text{STM} \to \text{LTM}) = 100\%$ to label the action.

We've also randomly dropped 20% of the input observations ($z_t$) to enhance the robustness of auto-regression. Among these, 10% of the input observations are replaced with a special mask token with trainable embeddings, while the remaining 10% are perturbed with Gaussian noise $\mathcal{N}(0, 1)$.

### A.4 Additional Results in Meta-Language Model

We test the ability of generation of the meta-language model on two rudimentary real-world language tasks: English vocabulary learning and basic mathematical operations (addition and subtraction), which are typical for primary school education. For the English vocabulary task, the vocabulary consists of alphabet characters and common punctuation marks; for the math task, it includes digits and mathematical operators. The input contexts for testing included 40 English words and simple arithmetic operations below 5. To discourage direct replication, we intentionally introduced errors into 15% of the words or equations. We analyzed the outputs of the MetaLM across two different context lengths, with the shorter contexts comprising only the first half of the longer contexts. The results, depicted in Fig. 16, show that MetaLM is capable of learning to memorize and correct both English vocabulary and mathematical equations, validating that its ICL capability is independent of vocabulary settings. Furthermore, the longer the context provided, the fewer errors appeared in the outputs, indicating a clear trend of improvement.

### A.5 Additional Results in Maze World Tasks

In Figure 17, We display trajectories generated by different policies on three randomly selected $35 \times 35$ tasks. Interestingly, we find that the main bottleneck of performance might arise from a lack of efficient exploration. Models with less context tend to explore less area. For instance, the random policy can only traverse a very small part of the maze. This observation could also explain the extraordinary challenge posed by the tasks: Traditional randomized exploration, a common technique in reinforcement learning, proves inadequate because it often revisits the same areas repeatedly without retaining information about which parts have been explored, thereby failing to cover broader regions effectively. Therefore, the agent must not only learn how to memorize paths and navigate but also how to explore more effectively, generating high-quality "training data" within its context to improve its future performance.

In Figure 18, we illustrate a qualitative evaluation of the full-context causal transformer's world modeling capabilities. Across the majority of instances investigated, a noticeable enhancement in future predicting is evident as the context length grows, suggesting that the model progressively adapts to the intricacies of the maze. This observation indicates the promising applicability of GPICL to the realm of adaptive embodied intelligence.

### A.6 Additional Diversity in the Maze World

It is worth noting that Maze World is a lightweight and fast engine with an easy-to-use Python API based on GYM (Brockman et al., 2016) for simulating navigation in various mazes. It runs smoothly at over 20 frames per second on a standard CPU. In addition to the aforementioned details, Maze World incorporates various

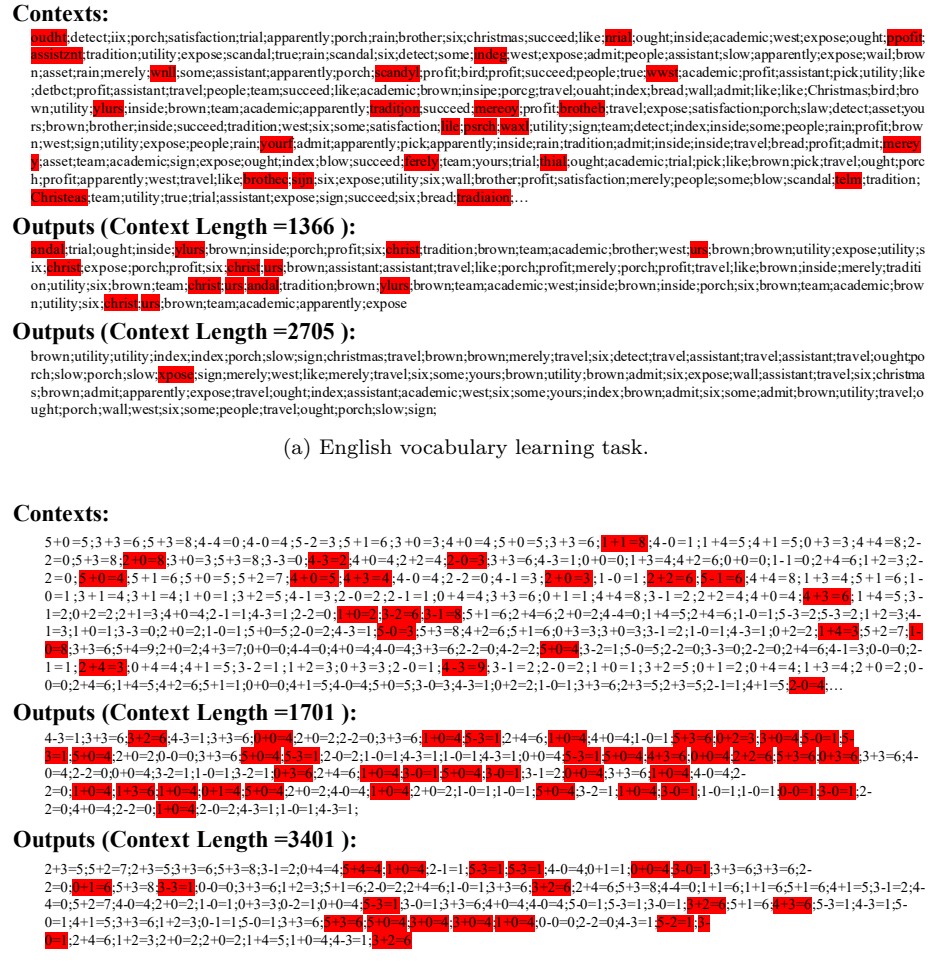

(a) English vocabulary learning task.

(b) Mathematical operation task.

Figure 16: Input contexts were designed such that 15% contained errors. All mistakes are highlighted with red squares. The vocabulary for both tasks was limited to 32 pre-defined tokens, and there was no pre-training or further tuning applied to either task. Notice that this setup tests the MetaLM's ability to adapt and correct based on in-context learning alone, without relying on prior knowledge or adjustments. The model under evaluation has **not been trained on any natural langauge corpora**. Additionally, the vocabulary is **randomly assigned** to the predefined 32 tokens prior to inference.

configurations and hyper-parameters that can be adjusted to create a wide range of tasks. For example, it offers different types of observation spaces and action spaces, including discrete and continuous actions, as well as 2D and 3D observations (Figure 19a). Beyond navigation tasks, we introduce more challenging survival tasks, where the rewards provided by the PNTs are static within an episode but initially unknown (can be negative, Figure 19b). The basic environments can also be modified in terms of the size of the basic cell (Figure 19c,19d), the height of the ceiling (Figure 19e,19f), the density of obstacles (Figure 19g,19h), and the range of view (Figure 19i,19j).

## A.7   Details of Task Configurations

In Table 2, we list the detailed settings for the tasks used in this paper. The rewards for reaching the target are by default related to the size of the maze. Additionally, we note that the density of obstacles controls the number of loops in the environments. We recommend using a density between 0.15 and 0.40. A density over 0.50 results in mazes without any loops, significantly reducing the difficulty of traversing the maze (or

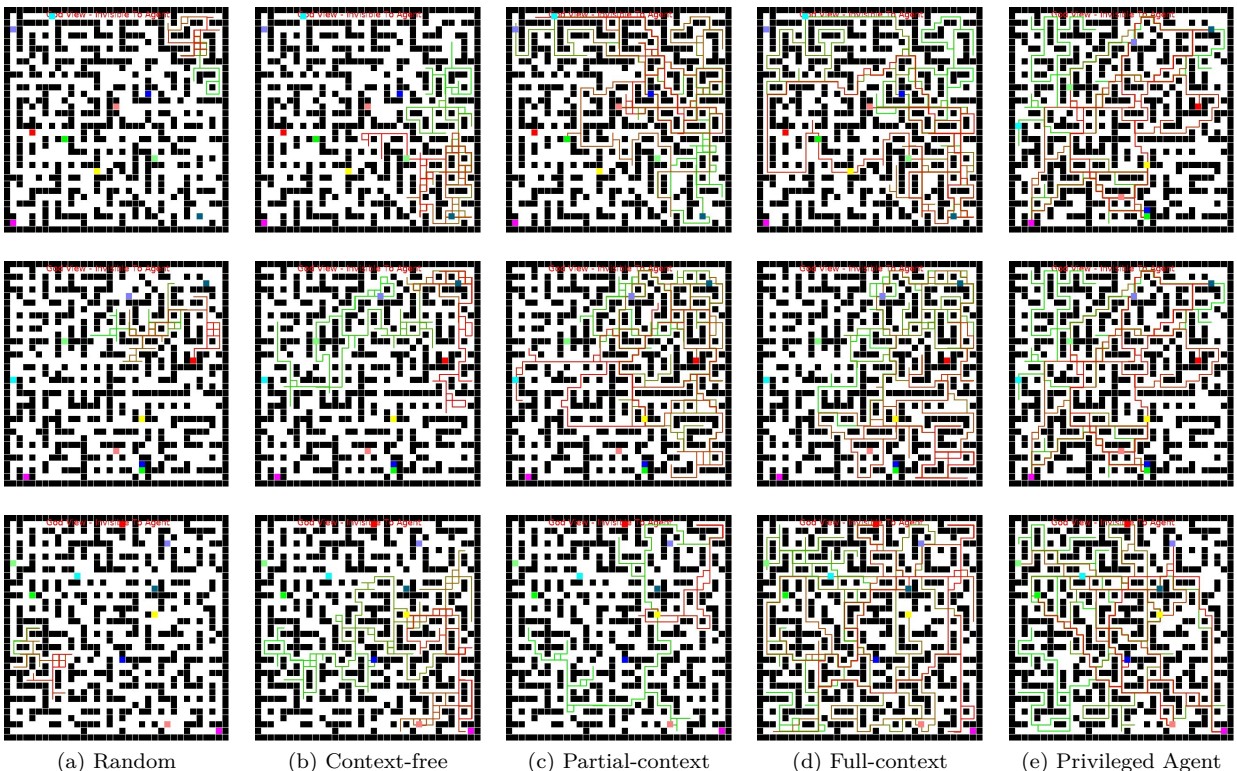

(a) Random     (b) Context-free     (c) Partial-context     (d) Full-context     (e) Privileged Agent

Figure 17: Performances of different methods in 3 randomly picked $35 \times 35$ tasks. Green trajectories indicate time steps $t$ close to 0, while red trajectories represent $t$ close to $T$. From left to right, the columns are generated by: (a) Random Policy (b) Context-free Model (c) Partial-context Model (d) Full-context Model (Small-sized) (c) Previleged agent with $p(\text{STM} \rightarrow \text{LTM}) = 100\%$.

performing exploration), as an agent can easily traverse the entire world by always going right (or left). We believe this might be a crucial reason why, in Morad et al. (2023), where the agent is only required to navigate to a single target, context-free models can outperform memory-augmented models.

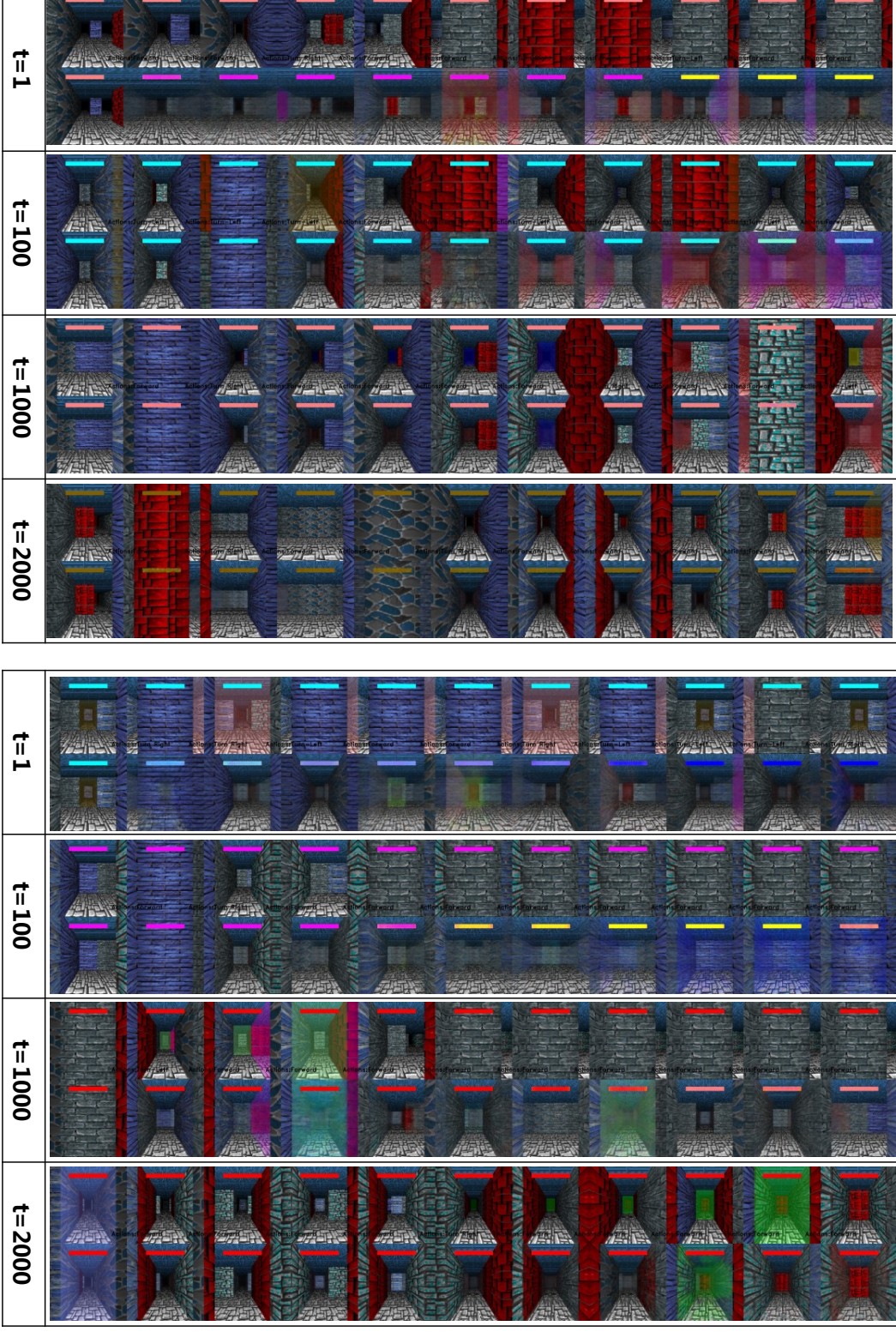

Figure 18: Cases regarding the in-context improvement of world modeling (with $15 \times 15$ mazes, small-sized full-context causal transformer). Each clip predicts 9 steps into the future, conditioned on a fixed sequence of actions, and based on varying lengths of context denoted by $t$. Each clip consists of two sections: the upper section displays the ground truth observations from the current step to 9 steps ahead (from left to right), while the bottom section shows the predicted future states up to 9 steps ahead.

Table 2: Setting of task configurations used in this paper

| **Meta-Langauge** | |
|---|---|
| Embedding size of the randomized langauge generator | 32 |
| Hidden size of the randomized language generator | 64 |
| Normalize factor for softmax of the randomized language generator ($\lambda$) | 5 |
| Number of the hidden parameters ($|\theta|$, $n = 2$) | 7K |
| Number of the hidden parameters ($|\theta|$, $n = 4$) | 11K |
| Number of the hidden parameters ($|\theta|$, $n = 8$) | 19K |
| **Maze Tasks** | |
| Density of obstacles | 0.36 |
| Number of Potential Navigation Targets (PNTs) | 10 |
| Resolution of observation | $128 \times 128$ |
| Range of view | 12.0 (default) |
| Size of basic cell | 2.0 (default) |
| Rewards of arriving target ($15 \times 15$) | 0.57 (default) |
| Rewards of arriving target ($25 \times 25$) | 1.24 (default) |
| Rewards of arriving target ($35 \times 35$) | 2.06 (default) |
| Height of ceiling | 3.2 (default) |
| Height of agents' view points | 1.6 (default) |

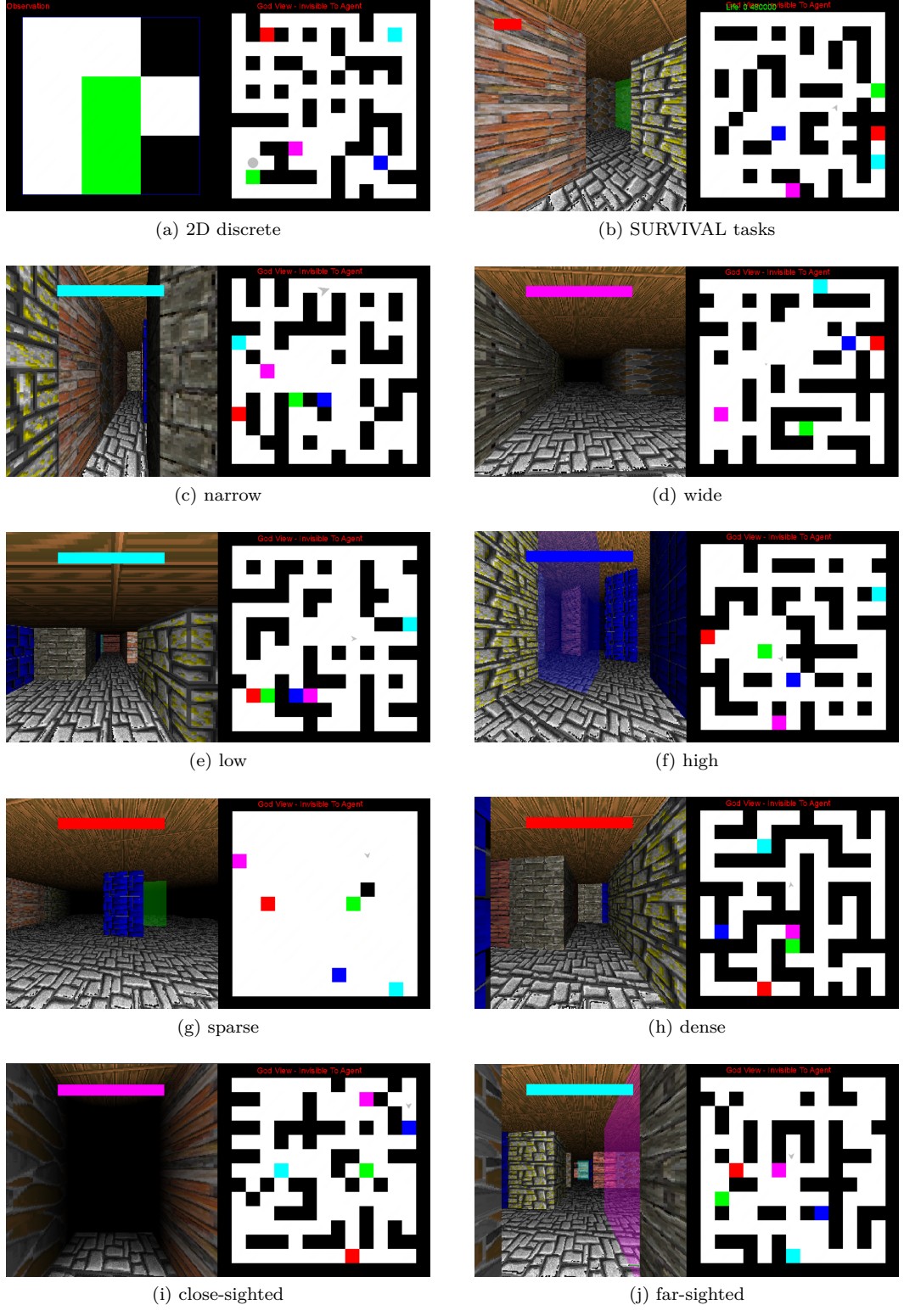

Figure 19: Additional variants by provided configurations in Maze World.

