# OpenReview forum: "Benchmarking General-Purpose In-Context Learning"
_TMLR — Rejected by TMLR_

### Review · Reviewer_FKYU · 2024-07-24

**Summary Of Contributions:**

The authors investigate in-context learning, the phenomenon where models are able to infer a task from part of their context and perform it later.

The goal is to define a benchmark suite of tasks to study the performance of in-context learning. Since the goal is to benchmark generic in-context learning, the authors want to use tasks that are different from the initial training mixture. The desired goal is to have a variety of tasks where 0-shot performance is very low, such that the model must infer from context how to perform the task.

The authors define further goals for the benchmark suite (long horizon, interactive, generative) and propose 2 sets of tasks: a meta-language task, where the model needs to learn a randomized but consistent language, and a maze world task where the agent needs to learn how to explore randomized mazes. Transformers of sizes 300k to 200M params are tried.

**Audience:**

Yes

**Claims And Evidence:**

Yes

**Requested Changes:**

No specific changes requested, but I do think the paper would be significantly stronger if it could argue or demonstrate an example of why these benchmarks with less inductive bias are needed, relative to existing language modelling datasets. Is there a way to demonstrate that the inductive bias in typical sequence benchmarks causes misleading results later?

In some sense, given the amount of time spent on the Maze World benchmark, it feels like the paper is more arguing for interactive, generative environments as benchmarks. If that's true, it's not very clear from the abstract. Mostly it just feels like an odd fit to have both an RL environment and a synthetic language task in the same paper, arguing that both are "in-context learning". It feels like it is just appealing to the lowest common denominator between the two, which makes the paper less interesting. Research papers ought to be opinionated.

**Strengths And Weaknesses:**

In-context learning of large sequence models is one of the the most useful behaviors and further understanding / benchmarking of those tasks could be important. However I feel like the tasks in this benchmark are still opinionated and I'm not sure they approach the 0 inductive bias goal, or measure things meaningfully different from other benchmarks.

The randomized language task, for example, is good. In this task, the goal is to match a fixed, randomized neural network's n-gram frequency. Given enough data, the model can learn to fit the patterns of the network, and show some ability for the model to predict the PG-19 project gutenberg test set. N-grams are a generic enough concept that it would be very hard to have an inductive bias on them.

On the other hand, I don't think the Maze World example is as good. The fact it is a 2d navigation task at all feels like an inductive bias to me. I am reminded of some of the criticisms of the procedurally generated environments from the Procgen Benchmark. Although they did try to make the games visually different, in the end they were still categorizable as 16 games. The Maze World task feels similarly, where the fact it is a randomly generated maze does not change that it is a maze navigation task.

---

> ### Author Response · Authors · 2024-07-24
> **Reply by the authors to Reviewer FKYU**
>
> We are grateful for the insightful feedback from the reviewer.
>
> We concur with the reviewer's observation that our proposed benchmarks, particularly "MazeWorld," exhibit significant inductive biases. In our paper, we advocate for GPICL to embody "Minimal Inductive Bias and Low Zero-Shot Capability." We realize that it is important to further clarify that Low Zero-Shot Capability is a consequence of minimal inductive bias during meta-training, but not necessarily vice versa. Specifically, by "Low Zero-Shot Capability," we mean that no model or agent can instantly adapt to a new task without adequate contextual information.
>
> We would like to highlight that "MazeWorld" is fundamentally distinct from the Procgen Benchmark (Cobbe et al., 2020). We first clarify that MazeWorld is a 3-D partially observable navigation benchmark (the 2D map is just used for analysis). In the Procgen Benchmark, each of the 16 categories contains thousands  of "levels," with each "level" (or namely task by our paper) being solvable through mainly zero-shot capability alone. For example, the "maze" category in Procgen, while it may have diverse  layouts just as "MazeWorld," the global map, its position and target is fully observable, meaning that models do not rely on context to solve any specific task. Furthermore, if a world model for mazes were trained in the Procgen environment, we would not expect to see an "In-Context Improvement of World Modeling." This is because an ideally trained model should be able to predict the next frame from the beginning of an episode, a feat that is not possible in "MazeWorld." In "MazeWorld," regardless of the number of parameters or the amount and the variety of tasks meta-trained, a model with less or no context cannot compete with those that have fewer parameters but more context from the specific task.
>
> To summarize briefly, while the tasks in the Procgen Benchmark primarily emphasize models' "zero-shot capability," "MazeWorld" focuses on "In-Context Learning with out much Prior Knowledge." But we need to admit that inductive bias do exist in MazeWorld, including the task definition itself and highly crafted observation and action space.
>
> We do not anticipate "MazeWorld" to serve as a universal benchmark for the RL or Embodied AI (EAI) community. However, we believe it offers a valuable benchmark for learning end-to-end Path Planning and "Simultaneous Localization and Mapping (SLAM)," which could be of interest to the robotics and autonomous navigation communities. Additionally, we believe "MazeWorld" a ideal testbeds for sequence encoding-decoding structures like Mamba, RWKV, xLSTM, Linear Attention, and others.
>
> The two benchmarks are linked by their shared characteristics of "Low Zero-Shot Capability, Generative Nature, Long In-Context Learning (ICL) Horizon (we claim that both benchmarks can be extend to nearly infinite context length by scaling up vocabulary, n-gram, and scale of mazes), and High ICL Potential (ideal long-term dependency and performance - context length relationship that most other benchmarks do not have)", which we believe to identify "GPICL". We acknowledge that the current interest group for both benchmarks may be limited. We also recognize that identifying Meta-RL benchmarks with "minimal or even zero inductive bias" would be a more significant achievement, but we find this task to be exceedingly challenging at present, and we will be working toward this direction in the future.

---

### Review · Reviewer_MM98 · 2024-07-26

**Summary Of Contributions:**

This paper describes two new benchmarks for general-purpose in-context learning (GPICL) and tests them with baseline models and approaches.

The first benchmark is for language modeling, and involves training on a collection of long sequences each generated by a different randomly initialized n-gram language models with varying context lengths (n=2…6). The test set consists of held out randomly-generated models, and English books. The baseline tested here is Transformer language models with moderate context lengths (4,096 tokens, if I understood correctly).

The second benchmark is for maze navigation in a 3-dimensional visual environment and randomly-generated mazes varying in size. A rule-based partial oracle is provided as a teacher model for imitation learning, allowing for RL-like behavior with a purely causal student model. The baseline evaluated here is an auto-encoder that predicts both the action and the next state, trained using DAGGER imitation learning.

**Audience:**

Yes

**Broader Impact Concerns:**

None.

**Claims And Evidence:**

Yes

**Requested Changes:**

I don’t really have any changes to request. My weaknesses above are high-level, philosophical and aspirational.

In case it is not apparent from my review, my expertise is in NLP and LLMs, not meta-learning or in-context learning. This work seems to need a few more iterations before it impacts my home field one way or the other, but I learned something from it and was inspired by it, and it is unlikely to cause harm from my perspective. I cannot in any way comment on whether it has left out any key citations or is any way harmful to the meta-learning community.

**Strengths And Weaknesses:**

Strengths:

- I find the notion of in-context learning a language very compelling, and I like this approach of learning from many different instances of randomly-generated Transformer models.
- The observations tying together generator-complexity, model scale and context length were very interesting. It would have been nice to have seen much longer n-gram lengths (and therefore higher complexity) in the generating models. Also, I wouldn’t call that section “scaling laws”, as no law is derived. I still like it, though.
- The inclusion of an English book task in the language modeling benchmark offsets almost all of my concerns about training and evaluating on synthetic data.
- The maze challenge seems solid, though perhaps less revolutionary, differentiated from what’s come before primarily by its 3D observation space and its diversity of layouts and tasks.

Weaknesses:

- The maze task seemed much less well-defined to me than the language modeling task. In writing my description above, I had a hard time finding key details, like the width and height in pixels of the visual observations.
- I’m not sure how I feel about the language benchmark. As mentioned, I love the task of in-context learning a language, but I’m not sure the benchmark should prescribe the training data. In a post-LLM world, we know that data is actually the most important variable by far, so it would be great if people could try to beat the provided baselines not by varying the learner (unlikely to work, in my opinion) but the data. Ideally, you’d want to set up a benchmark’s test set and evaluation such that standard LLM training data will fail due to devoting too much capacity to zero-shot performance, and not enough to ICL performance. That way we could race standard LLMs against this new training data proposal.
- Just learning the next-token prediction task with ICL seems like only a first, tiny step, highly related to memorizing frequent tokens. It would be amazing to also learn more complex language tasks.
- Similar to my concerns about the language modeling task, it isn’t clear to what extent the maze task is defined by its training data or just by its test sets. The text is written in such a way to imply that the imitation-learning technique + partial oracle teacher used here is not part of the task, but part of the baseline, but the privileged agents are described in the problem setting section, muddying the waters.
- The bulk of the front matter and the related work section is devoted to helping people understand GPICL. I’m afraid, even having read the paper and mostly understood these benchmarks, I could not confidently draw the line between ICL and GPICL. It seems quite fuzzy. This is not helped by the fact that the two benchmarks are only related by having randomly generated training data, and are not even addressed by the same baseline learning method.

---

> ### Author Response · Authors · 2024-08-07
> **Reply by the authors to Reviewer MM98**
>
> # Does GPICL and ICL has a clear boundary?
>
> To clarify, the distinction between GPICL and ICL is fundamentally determined by the diversity and quantity of tasks used in training, rather than the specific model architecture or the meta-training process itself.
> In theory, we propose that GPICL should have a lower inductive bias, which implies a reduced ability to perform well without training (lower zero-shot capability) but a greater potential for ICL (higher ICL potential). It should also have a longer ICL horizon, as we explained in Section 1. However, we acknowledge that there is no precise threshold that definitively distinguishes ICL from GPICL.
> To illustrate this, let's consider a hypothetical experiment: Suppose we begin meta-training with a task set that is relatively homogeneous, containing a sufficient number of tasks but lacking in diversity. Under these conditions, the meta-trained model is expected to excel in in-distribution (ID) tasks, which might be classified as “zero-shot capability”,  but have a limited ICL potential in addressing out-of-distribution (OOD) tasks. As we progressively enhance the diversity of tasks in our training set, we hypothesize that the model's “zero-shot capability” will diminish, and the ICL horizon increases as the diversity of tasks stops the model from “memorizing all possible tasks” and make “fast identification of seen task” , while its ICL potential also increases as the OOD task gradually become ID as a result of the increase of diversity of training tasks.
>
> # About GPICL and the standard LLMs.
>
> We acknowledge that our current efforts represent a tiny step in a much larger map. The GPICL we emphasized introduces a learning pipeline that diverges significantly from the mainstream LLMs in use today. Our proposal does not prioritize benchmarking over data; rather, it suggests that data diversity is paramount, even if it introduces additional biases. We advocate for a reevaluation of current benchmarks, as they may not accurately reflect the capabilities of models.
> 1. Many existing 'long-term dependency' natural language benchmarks do not genuinely test for long-term dependencies, as we have discussed in the 'Related Works' section. Consequently, techniques like Mixture-of-Experts, which focus more on zero-shot capabilities than ICL capabilities, can score highly in evaluations. This is rooted in the fact that if the training data lacks true diversity, models tend to rely more on 'parameter memory'—memorizing the training data—rather than 'context memory'—understanding the context within the data.  It is also frustrating that emphasizing “zero-shot capability” rather than “In-context learning” could be more valuable practically. We believe that if our ultimate goal is AGI, it is essential to first create datasets and sequences that encourage the learning agent to continually learn anew from a blank slate.
> 2. Synthetic benchmarks designed to test long-term dependencies, such as 'Needle-In-The-Sea,' resemble information retrieval tasks more closely. Recently we have just observed that memory-based sequence encoders, like LSTM and Mamba, perform significantly worse than Transformers in these tasks. However, they can perform almost as well in our meta-language benchmarks. It is a common understanding in human intelligence that we are not adept at memorizing long but unstructured information. Yet, we excel at systematically learning and refining specific skills in a relatively long time. We think it is premature to conclude that memory-based sequence encoders are inadequate based on the test results on those benchmarks.
>
> # Baseline learning methods (for meta-training) are different.
>
> The MazeWorld benchmark aims to shed light to the development of foundational models for Reinforcement Learning and Embodied AI. However, we encountered significant challenges in devising a benchmark that matches the complexity and scope of meta-language tasks. Our proposal does not advocate for a 'one-size-fits-all' meta-training approach for GPICL. This includes not only self-supervised learning for language and world models but also imitation learning for policy models and reinforcement learning strategies.
> We think that further research is necessary to uncover more effective meta-training methodologies. At this stage, our findings indicate that only through a combination of imitation learning and self-supervised learning can we effectively meta-train across such a broad spectrum of tasks and sequences.

---

### Review · Reviewer_Hh5j · 2024-07-30

**Summary Of Contributions:**

- The authors study extending in-context learning (ICL) to address a broader range of tasks with an extended learning horizon and higher improvement potential, which they call General Purpose In-Context Learning (GPICL).

- They introduce two synthetic benchmarks designed to train and evaluate GPICL:
  - Meta-Language: Generates any number of randomized yet consistent "languages" (basically n-gram models with varying parameters) to test a model's ability to learn a new language from scratch through ICL, without exposure to natural language corpora.
  - Maze World: Generates randomized mazes to test a model's ability to explore, memorize surroundings, identify targets and plan routes solely through context.

- The benchmarks are characterized by significant task variance to minimize inductive bias during meta-training. The tasks promote long-horizon ICL through continuous generation and interaction.

- Experiments on baseline transformer models demonstrate the feasibility of meta-training with minimal inductive bias and ICL across language modeling, decision-making, and world modeling domains.

**Audience:**

Yes

**Claims And Evidence:**

Yes

**Requested Changes:**

Please address all the weaknesses

**Strengths And Weaknesses:**

Strengths
- The authors make a good attempt to provide two interesting general purpose benchmarks for GPICL
- While the benchmarks are somewhat limited (eg n-gram for modeling natural languages), they have good diversity to minimize inductive bias.
- Experimental results give an indication Transformer architectures can achieve high performance on both benchmarks. They also find that the performance of the models is more strongly correlated with the length of the context than with the number of parameters, suggesting that increasing the context length may be more beneficial for GPICL than increasing the model size.

Weaknesses
- *grammar:*  The paper is very poorly written (in terms of grammar); consider properly proofreading it. Figure labels are practically unreadable, considers increasing the font size.
- *Lack of comparison to existing benchmarks:* The authors do not compare the performance of the Meta-Language and Maze World benchmarks to existing benchmarks for in-context learning. This makes it difficult to assess the relative difficulty of the benchmarks and to determine whether they provide a more comprehensive evaluation of GPICL capabilities.
- *Limited scope:* The Meta-Language and Maze World benchmarks only assess two aspects of GPICL: translation and navigation. There are many other aspects of GPICL that could be assessed, such as the ability to perform reasoning, or answer questions.
- *Lack of theoretical foundation:* The authors do not provide a theoretical foundation for the Meta-Language and Maze World benchmarks. They do not explain why these benchmarks specifically as opposed to other synthetic/realworld meta-learning benchmarks (either previously existing ones or creating new ones) are important for assessing GPICL capabilities,

---

> ### Author Response · Authors · 2024-08-07
> **Reviewer Hh5j**
>
> Regarding the phrases and representations: We will enhance the document by incorporating additional proofreading and by enlarging the font size within figures to improve readability and clarity.
>
> Lack of comparison to existing benchmarks: We believe that comparing different benchmarks may not be particularly meaningful. Our primary focus is on introducing new benchmarks and highlighting crucial aspects for evaluating GPICL. While we considered enhancing our study by including a broader range of baseline models for comparison—such as various recurrent structures (RNN, LSTM, xLSM) and efficient transformer models (Transformer-XL, Block Recurrence, Mamba)—currently we think that this was less central to our main objective. The critical points we aim to address are the potential implications of increasing the diversity and quantity of tasks during meta-training, specifically the potential increase in Inductive Bias of Language Models (ICL) and the possible decrease in zero-shot capabilities.

---

### Author Response · Authors · 2024-08-21
**New Version Updated**

We revise the paper and upload an updated version.
1. Revise Phrases and Sentences: Reword certain phrases and sentences, and claims to make them more grounded. For instance, place emphasis on "Long-term In-Context Dependency" and "High In-Context Improvement Potential" instead of “Minimum Inductive Biases”
2. Add a New Section: Introduce a new section, titled "Section 4: Emergence of Generalizability," which includes a series of ablation studies. In these studies, we meta-train with varying numbers of tasks and demonstrate that an increased number of tasks leads to an extension of the ICL Horizon, a reduction in zero-shot performance, and an enhancement of generalizability.
3. Improve Figure Clarity: Enhance the clarity of the figures, particularly by making the font more legible. Specifically, for figures depicting the performance of the world model, use context length as the horizontal axis to more clearly illustrate the in-context improvement of world modeling.
In the revised version we indicate the significant revisions using a blue color.

---

### Decision · Action_Editor_iQW6 · 2024-09-07

**Recommendation:** Reject

**Comment:**

The contributions of this work are not fully substantiated, as no previous benchmarks have been evaluated within the GPICL framework. Furthermore, during the response period, the authors did not address this concern raised by two reviewers. Consequently, this paper does not meet the criteria and standards of TMLR.

**Audience:**

There should be a wide community of audience for this work.

**Claims And Evidence:**

This paper introduces two new benchmarks for general-purpose in-context learning (GPICL) and evaluates them using baseline models and methods—one for language modeling and the other for maze navigation. While all reviewers recognize the importance of developing benchmarks to enhance our understanding of in-context learning, a primary concern, raised by nearly all three reviewers, is whether these two benchmarks provide any distinct advantages or differences compared to existing language modeling and visual navigation benchmarks in addressing the motivations of this work within the GPICL framework. This concern is particularly pertinent given that the new benchmarks have not been directly compared with previous ones. As noted by Reviewer MM98, the value of these benchmarks should not be assessed solely based on their performance with different models; it is also crucial to consider whether these benchmarks pose unique challenges to LLMs, as highlighted by Reviewer Hh5j, in comparison to other existing benchmarks.

**Resubmission Of Major Revision:**

The authors may consider submitting a major revision at a later time.